# Deep Learning-Based Synthesized View Quality Enhancement with DIBR Distortion Mask Prediction Using Synthetic Images

**DOI:** 10.3390/s22218127

**Published:** 2022-10-24

**Authors:** Huan Zhang, Jiangzhong Cao, Dongsheng Zheng, Ximei Yao, Bingo Wing-Kuen Ling

**Affiliations:** School of Information Engineering, Guangdong University of Technology, Guangzhou 510006, China

**Keywords:** synthesized view, quality enhancement, synthetic images, data augmentation

## Abstract

Recently, deep learning-based image quality enhancement models have been proposed to improve the perceptual quality of distorted synthesized views impaired by compression and the Depth Image-Based Rendering (DIBR) process in a multi-view video system. However, due to the lack of Multi-view Video plus Depth (MVD) data, the training data for quality enhancement models is small, which limits the performance and progress of these models. Augmenting the training data to enhance the synthesized view quality enhancement (SVQE) models is a feasible solution. In this paper, a deep learning-based SVQE model using more synthetic synthesized view images (SVIs) is suggested. To simulate the irregular geometric displacement of DIBR distortion, a random irregular polygon-based SVI synthesis method is proposed based on existing massive RGB/RGBD data, and a synthetic synthesized view database is constructed, which includes synthetic SVIs and the DIBR distortion mask. Moreover, to further guide the SVQE models to focus more precisely on DIBR distortion, a DIBR distortion mask prediction network which could predict the position and variance of DIBR distortion is embedded into the SVQE models. The experimental results on public MVD sequences demonstrate that the PSNR performance of the existing SVQE models, e.g., DnCNN, NAFNet, and TSAN, pre-trained on NYU-based synthetic SVIs could be greatly promoted by 0.51-, 0.36-, and 0.26 dB on average, respectively, while the MPPSNRr performance could also be elevated by 0.86, 0.25, and 0.24 on average, respectively. In addition, by introducing the DIBR distortion mask prediction network, the SVI quality obtained by the DnCNN and NAFNet pre-trained on NYU-based synthetic SVIs could be further enhanced by 0.02- and 0.03 dB on average in terms of the PSNR and 0.004 and 0.121 on average in terms of the MPPSNRr.

## 1. Introduction

With the development of video capture and display technology, a 3D video system could provide people with a more and more immersive and realistic sensation, such as six-degrees-of-freedom (6DoF) video, which is close to the viewing experience of people interacting with the real world. However, along with the sensory impact of an immersive visual experience, the associate data volume has increased dozens of times, which has brought great challenges to the collection, storage, and transmission of virtual reality. In order to alleviate the pressure of storage and bandwidth, it is necessary to increase the compression ratio or use more sparse viewpoints to synthesize the virtual view/synthesized view. These processes will inevitably bring distortion to the video and damage the visual perception quality of users. To improve users’ visual experience, it is necessary to enhance the image quality of the synthesized view.

In a synthesized view image (SVI), there exists compression distortion and synthesis distortion caused by the DIBR process, and it is difficult for conventional image denoising and restoration models to deal with or eliminate these distortions due to their complexity. Learning-based image denoising and restoration models have been proved to be effective in dealing with such distortion, better for their powerful learning ability. In SynVD-Net [1], the compression and DIBR distortion elimination in the synthesized video was modeled as a perceptual video denoising problem, and a derived perceptual loss was derived and integrated with image denoising/restoration models, e.g., DnCNN [2], a U-shape sub-network in CBDNet [3], and a Residual Dense Network (RDN) [4], to enhance the perceptual quality. In [5], a Two-Stream Attention Network (TSAN) was proposed by combining a global stream which extracts the global context information and a local stream which extracts the local variance information. Following [5], a Residual Distillation Enhanced Network (RDEN)-guided lightweight synthesized video quality enhancement (SVQE) method [6] was proposed which claims to address the huge complexities and effectively deal with the distortion in the synthesized view. However, the existing Multi-view Video plus Depth (MVD) database has few sequences; thus, few (insufficient) noisy/clean sample pairs with various content are available for learning. This may hinder the ability of SVQE models, and it may be unable to fairly evaluate the capabilities of SVQE models.

How to improve the SVQE model performance with limited training data remains a non-trivial problem. There are different ways to improve the performance, such as data augmentation, model structure regularization, pre-training, transfer learning, and semi-supervised learning, among which the latter three learning-based techniques are usually conducted with data augmentation. Because massive natural RGB or RGBD images are easily accessible or available, in this paper, these data are utilized to simulate DIBR distortion and construct the synthetic SVIs, based on which the SVQE models could be first pre-trained and then fine-tuned on limited MVD data. In addition, in order to better improve the quality of SVIs, the human perception toward the virtual view is considered by embedding the DIBR distortion mask prediction network which could predict the position of the DIBR distortion into the SVQE models. The major contributions of this paper are threefold.

A transfer learning-based scheme for the SVQE task is proposed, in which the SVQE model is first pre-trained on a synthetic synthesized view database, then fine-tuned on the MVD database;A synthetic synthesized view database is constructed in which a specific data synthesis method based on the random irregular polygon generation method simulating the special characteristics of SVI distortion is proposed, which has been validated on well-known state-of-the-art denoising or SVQE models on public RGB/RGBD databases;A sub-network is employed to predict the DIBR distortion mask and embedded with SVQE models using synthetic SVIs. The attempt of explicitly introducing the DIBR distortion position information is proved to be effective in elevating the performance of SVQE models.

The remainder of this paper is organized as follows. Section 2 reviews the related work. The motivation and the proposed random polygon-based DIBR simulation and the DIBR mask prediction methods are proposed in Section 3. Section 4 describes the datasets preparation. Section 5 presents the substantial experimental results and detailed analyses. Section 6 concludes the paper.

## 2. Related Work

In this section, the development of image denoising methods and SVQE models is first reviewed, followed by a brief introduction of image data augmentation and data synthesis, and ended with recent research works about distortion mask prediction in image denoising or restoration.

### 2.1. SVQE Models

Image denoising or restoration is a classical image processing low-level task, which attracts an enduring passion from academy and industry. The attributions of state-of-the-art image denoising and SVQE methods are listed in Table 1. To begin with hand-crafted feature-based methods, NLM [7] and BM3D [8] are the most classical conventional image denoising methods which utilize the non-local self-similarity in images or image sparsity in the transform domain. Recently, with the development of deep learning, numerous image denoising and restoration methods have sprung up like bamboo shoots. For example, DnCNN [2], FFDNet [9], CBDNet [3], RDN [4], and NAFNet [10] were proposed successively with increasing denoising ability. These methods were initially proposed for uniform noise, such as Gaussian noise and real image capturing noise. With the great success of a transformer which has been applied into various computer vision tasks, transformer-based networks, such as Restormer [11] and SwinIR [12], have been proposed for low-level image processing tasks, e.g., real image denoising and image super-resolution. These transformer-based image restoration networks could model long-range relationships in images, which are beneficial to image restoration, especially for DIBR structure distortion. However, the disadvantages are that they consume large computational resources and need a large amount of data. In this paper, we mainly focus on CNN-based SVQE models.

For compression distortion caused by codec, VRCNN [13] and other compression methods [14,15] were proposed to deal with the blocking artifacts and texture blur caused by the compression. In the MVD-based 3D video system, a virtual view is synthesized by a compressed texture and depth video through the DIBR process, which includes the uniform compression distortion (mainly transferring from texture images) and irregular synthesis distortion (mainly originating from the DIBR process and distorted depth images). To improve the perceptual quality of SVIs during compression, Zhu et al. [16] proposed a network which was adapted from a DnCNN network and utilized the neighboring view information to enhance the reference synthesized view and refine the synthesized view obtained from compressed texture and depth video. Later, Pan et al. proposed the methods TSAN [5] and RDEN [6] to improve the SVI quality. To improve the perceptual synthesized video quality, SynVD-Net [1] was proposed by deriving a CNN-friendly loss from the perceptual synthesized video quality metric to reduce the flicker distortion. These SVQE models could better enhance the SVI quality. However, due to the limited MVD data, the potential of SVQE models may not be fully excavated.

**Table 1 sensors-22-08127-t001:** The attributions of state-of-the-art image denoising and SVQE methods.

Methods	Hand-Crafted	CNN-Based	Transformer-Based	General 2D Images	Synthesized Views	Main Noise Types
NLM [7], BM3D [8]	✔			✔		Gaussian noise
DnCNN [2], FFDNet [9], CBDNet [3], RDN [4], NAFNet [10]		✔		✔		Gaussian noise, blur, real image noise, low resolution
Restormer [11], SwinIR [12]			✔	✔		
VRCNN [13], Zhu et al. [16], TSAN [5], RDEN [6], SynVD-Net [1]		✔			✔	Compression distortion, DIBR distortion

### 2.2. Image Data Augmentation and Data Synthesis

Image data augmentation has been widely used in learning-based computer vision tasks, which includes basic (classic and typical) [17,18,19,20] and deep learning-based [21,22,23,24] data augmentation methods. Basically, the basic data augmentation methods can be categorized as data warping [17], e.g., geometric and color transformation [18], mixing image [19,20], random erasing, and so on. These augmentation methods use oversampling or data warping to preserve the label. The deep learning-based data augmentation methods can be classified as GAN-based [21], neural-style transfer [22], adversary training [23,24], and so on. The above data augmentation methods are general and could partially improve the performance of related image processing tasks. Another common way for using limited data in computer vision application is to use transfer learning to pre-train a model on a large-scale external database or use domain adaptation methods. However, often there is a certain feature gap between an external database or a pre-trained model and the downstream specific tasks [25].

Recently, the domain-specific data synthesis methods which could utilize the strong prior knowledge of target images are used in many tasks and have demonstrated its effectiveness in real image denoising [3], rain removal [26], shadow removal [27,28], and other tasks [29,30]. Because real MVD data is limited, it is difficult to obtain real multi-view data and the associated information, e.g., camera parameters and depth values, thus making synthesizing an SVI thorny. To tackle this issue, a DIBR distortion simulation has been proposed in some image quality assessment (IQA) studies for 3D synthesized images. In [31], a DIBR distortion simulation was proposed so as to predict the DIBR-synthesized image quality without a real time-consuming DIBR process. However, the virtual view synthesis method utilizes wavelet transform to mix high-frequency signals near the texture and depth edges and could not simulate the random geometric displacement distortion caused by the depth value error. In [32], a DIBR distortion simulation was realized by a hand-crafted and GAN-based method to solve the data shortage problem. The DIBR distortion synthesized by a GAN may not well match the distribution of DIBR distortion and it needs large data and data-labeling to train, which is troublesome. In this paper, we aim to propose a simple data synthesis method for DIBR distortion simulation.

### 2.3. Distortion Mask Prediction

In some image restoration tasks, e.g., real image denoising [3], de-raining [33], or shadow removal [28,34], the image restoration task is explicitly or implicitly divided into two tasks, i.e., distortion mask (or labels) estimation and image restoration/denoising. In [3], a noise estimation sub-network is embedded into an image denoising framework. In [33], the de-raining network is composed of a rain density-aware network for the rain density label prediction and de-rain network and were jointly learned. In [28], a novel Dual Hierarchical Aggregation Network (DHAN) was proposed which can simultaneously output a shadow mask and a shadow-erased image using the GAN-synthesized shadow images. In [34], a distortion localization network is integrated with an image restoration network to handle spatially varying distortion. These works [3,28,33] all used self-created synthetic images except that pseudo distortion labels were used in [34].

## 3. Method

In this section, the motivation is first illustrated. Pipeline of DIBR distortion simulation is then described, in which different kinds of local noise are compared and the proposed random irregular polygon-based DIBR distortion generation method is introduced. Thus, synthetic databases could be constructed with synthetic SVIs and corresponding DIBR distortion masks. Last, the DIBR distortion mask prediction sub-network is introduced and integrated with SVQE models based on the constructed synthetic SVIs. The definitions of key variables and acronyms used in this section are listed in Table 2.

### 3.1. Motivation

Nowadays, the Internet is abundant in high-quality specially constructed image databases or user uploaded images/videos. Thus, it is easy to obtain enough data to conduct the learning task, e.g., compressed image quality enhancement (denoted as task $T_{s}$), by collecting original RGB images and producing their corresponding compressed images by compression tools. Suppose a domain D which consists of data/feature space χ and a marginal probability distribution P(X) [35], where $X=\{x_{1},x_{2},\ldots ,x_{n}\}\in \chi$. However, in an MVD video system, insufficient data are available to recover high-quality from distorted synthesized images, i.e., synthesized view quality enhancement (denoted as learning task $T_{t}$), which may weaken the performance of SVQE models. Confronted with such situation that abundant image pairs $D_{s}=\left\{\left(x_{s_{1}},y_{s_{1}}\right),\left(x_{s_{2}},y_{s_{2}}\right),\ldots ,\left(x_{s_{n}},y_{s_{n}}\right)\right\}$ (ground truth/distorted images) could be collected for $T_{s}$ but limited image pairs $D_{t}=\left\{\left(x_{t_{1}},y_{t_{1}}\right),\left(x_{t_{2}},y_{t_{2}}\right),\ldots ,\left(x_{t_{n}},y_{t_{n}}\right)\right\}$ could be gathered for task $T_{t}$, it may naturally occur to people that transfer learning could be utilized to transfer from $T_{s}$ to $T_{t}$, as shown in Figure 1a. However, due to the discrepancy between $D_{s}$ and $D_{t}$, the knowledge that could be learned and transferred from task $T_{s}$ may only be compression distortion elimination knowledge, which may be suboptimal when applied on task $T_{t}$. In addition, the compression distortion is relatively regular, while DIBR distortion in SVI is more irregular and hard to handle.

To break the gap between domain $D_{s}$ and $D_{t}$, and make better use of the big data in domain $D_{s}$, a method is proposed to generate the synthetic noise simulating the DIBR distortion, aiming that the knowledge of synthetic noise distribution could be approaching true noise distribution in synthesized images, and thus could effectively utilize the massive data in domain $D_{t}$. As shown in Figure 1b, DIBR distortion simulation module is introduced after image compression, and synthetic synthesized images are thus generated accordingly.

### 3.2. DIBR Distortion Simulation

Figure 2 shows the pipeline of DIBR distortion simulation. Original images from NYU [36] and DIV2K [37] databases (public RGB/RGBD databases) are first compressed by using codec with given Quantization Parameter (QP) parameter. The associated depth images of the compressed images are available for RGBD images or could be generated by mono depth estimation methods [38,39]. Then, the DIBR distortion will be generated along the depth edges because depth edges are assumed to be the most possible areas where DIBR distortion resides. Next, the proposed random irregular polygon-based DIBR distortion generation method is employed on the compressed RGB/RGBD data. In this way, the synthetic synthesized view database is constructed, which includes synthetic synthesized images and corresponding DIBR distortion mask.

### 3.3. Different Local Noise Comparison and Proposed Random Irregular Polygon-Based DIBR Distortion Generation

Figure 3a demonstrates the SVI with DIBR distortion of sequences Lovebird1 and Balloons, such as cracker, fragment, and irregular geometric displacement along the edges of objects. To investigate which kind of distortion resembles the DIBR distortion more, three different noise patterns, e.g., Gaussian noise, speckle noise, and patch shuffle-based noise, are compared. Gaussian noise is a well-known noise with normal distribution. Speckle noise is a type of granular noise which often exists in medical ultrasound images and synthetic aperture radar (SAR) images. Patch shuffle [40] is a method to randomly shuffle the pixels in a local patch of images or feature maps during training which is used to regularize the training of classification-related CNN models. Taking the DIBR distortion simulation effects for Lovebird1 as example, as shown in Figure 3, different synthetic SVIs are obtained by adding compressed neighboring captured views with Gaussian noise, speckle noise, and patch shuffle-based noise along the areas with strongly discontinuous depth, respectively. The real SVI is listed as anchor. Denote the captured view as I, then the synthetic synthesized view by the random noise can be written as
(1)$$I_{syn}=\left(1-M\right)⊙I+\frac{M⊙(I+I_{\delta })}{2},$$
where $I_{syn}$ denotes the synthetic SVI, *I* denotes the compressed captured view images, ***1*** denotes the matrix with all elements as 1, *M* denotes the mask area corresponding to the detected strong depth edges, ⊙ denotes dot product, and $I_{\delta }$ denotes the images added with random noise, i.e., Gaussian noise, speckle noise, or the patch-shuffled version of *I*. It could be observed that $I_{syn}$ synthesized by Gaussian noise and speckle noise are not very visually resembling synthesis distortion, and $I_{syn}$ synthesized by patch-based noise exhibits similar behaviors a little in the way that the pixels in a local patch appear as disorderly and irregular.

SVI with DIBR distortion can be viewed as the tiny movement of textures within random polygon area along the depth transition area. To better simulate the irregular geometric distortion, in this section, a simple random polygon generation method which could control irregularity and spikiness will be introduced as follows. A random polygon generation method could be found in [41]. Following the method [42], to generate a random polygon, a random set of points with angularly sorted order would be first generated; then, the vertices would be connected based on the order. First, given a center point *P*, a group of points would be sampled on a circle around point *P*. Random noise is added by varying the angular spacing between sequential points and the radial distance of each point from the center. The process can be formulated as
(2)$$\left\{\begin{array}{c} \theta_{i}=\theta_{i-1}+\frac{1}{k}▵\theta_{i}\\ ▵\theta_{i}=U\left(\frac{2\pi }{n}-\epsilon ,\frac{2\pi }{n}+\epsilon \right),\\ k=\sum ▵\theta_{i}/\pi \\ ri=clip(N(R,),0,R) \end{array}\right.$$
where $\theta_{i}$ and $r_{i}$ represent the angle and radius between the *i*-th point and assumed center point, respectively. $▵\theta_{i}$ denotes the random variable controlling angular space between sequential points, which is subject to a uniform distribution featured by the smallest value $\frac{2\pi }{n}-\epsilon$ and largest value $\frac{2\pi }{n}+\epsilon$, where *n* denotes the number of vertices. Moreover, $r_{i}$ is subject to Gaussian distribution with a given radius *R* as mean value and σ as the variance. *R* could be used to adjust the magnitude of the generated polygon. ϵ could be used to adjust the irregularity of the generated polygon by controlling the angular variance degree through the interval size of *U*. σ could be used to adjust the spikiness of the generated polygon by controlling the radius variance through the normal distribution. Large ϵ and σ indicates strong irregularity and spikiness, and vice versa, which can be shown in Figure 4.

Thus, the synthetic SVI composed by the proposed random polygon noise can be obtained as
(3)$$\left\{\begin{array}{c} I_{syn}=\left(1-M\right)⊙I+\frac{M⊙(I+I_{sh})}{2},\\ I_{sh}\left(\psi \right)=I\left(\psi +\eta \right) \end{array}\right.$$
where ψ denotes the vertices set located in a local region generated by the random polygon method, η denotes a random vector for all points of ψ to be bodily shifted in $I_{sh}$. $I_{sh}$ is fused with *I* in the strong depth regions. In Figure 3f,j, it can be observed that the DIBR distortion generated by the activity of textures within random polygon area along the edges resembles the distortion visually.

### 3.4. DIBR Distortion Mask Prediction Network Embedding

Existing IQA models for SVI demonstrate that DIBR distortion position determination is the key procedure for quality assessment [43,44], which hints that knowing and paying more attention to DIBR distortion position may elevate SVQE models in enhancing SVI quality. Therefore, how to incorporate the DIBR distortion position into SVQE models becomes a new issue. The intuitive way is directly integrating DIBR distortion position with distorted image as a whole input. Figure 5a shows the sketch map of this way. It could be validated by experiment in Section 4 that knowing DIBR distortion position is helpful for synthesized image quality enhancement. However, the ground truth DIBR distortion position is often not known, so the position has to be detected or estimated. Inspired by de-raining [33] and shadow removal [28,34], SVI quality enhancement could be regarded as two tasks, i.e., DIBR distortion mask estimation and image restoration/denoising. Reviewing these works, there are three main possible ways to group mask estimation and image restoration task network, i.e., successive (series) network, parallel network (multi-task), parallel interactive network. The sketch map of these ways is demonstrated in Figure 5b–d. In addition to different organization or design of networks, attention mechanism, such as spatial attention [5], self-attention [10], or non-local attention [45], is also considered in existing denoising or restoration networks. In this work, we mainly focus on networks which explicitly combine the DIBR mask prediction and DIBR distortion elimination and mainly test the successive (series) network shown in Figure 5b.

## 4. Datasets Preparation

Two datasets, i.e., the RGBD database NYU Depth Dataset V2 [36] and the RGB database DIV2K [37], were employed for synthetic SVI database construction for the pre-training of SVQE models. MVD dataset from SIAT Synthesized Video Quality Database [46] was used as the benchmark dataset for SVQE.

NYU-based synthetic SVIs: NYU Depth Dataset V2 consists of RGB and raw depth images from various indoor scenarios captured by Microsoft kinect, which are originally proposed for image segmentation and depth estimation. The database is comprised of 1449 labeled datasets and 407,024 unlabeled datasets. In our experiment, only 1449 labeled datasets of aligned RGB and depth images are employed. The resolution of the images is 640 × 480. The images were compressed by X264 with QP, which was set as 35, an intermediate distortion level. The NYU-based synthetic SVIs were generated through Equations (2) and (3) on Y-component of compressed images.

DIV2K-based synthetic SVIs: DIV2K dataset consists of 1000 2K resolution RGB images with various content which was proposed for super-resolution. In our experiment, 750 images were employed for training. The compression and DIBR distortion generation procedures are the same as that in NYU Depth Dataset V2, and the QP was set as 45 for the high-resolution images, because QP 35 is not noticeable for DIV2K dataset.

Example images of constructed synthetic SVIs based on NYU Depth Dataset V2 and DIV2K are shown in Figure 6.

MVD: MVD dataset is the same as that in [1], and it includes 12 common MVD sequences with a variety of content. Selected reference views were compressed and then used to synthesize an intermediate view. Note 3DV-ATM v10.0 software [47] was used for compression and VSRS-1D-Fast software [48] was utilized for the reference views to render the intermediate virtual view. In experiments, five sequences were selected in training, and the left seven sequences were used in testing. The testing sequences are denoted as Seqs-H.264 for simplicity. In our test, we only trained and tested on the intermediate distortion level, and 10 or 21 images were collected from the distorted video, which are 94 training frames in total. The detailed information about sequences, view resolution, reference and rendered views, and compression parameter pairs ($QP_{t},QP_{d}$) for reference views of texture and depth videos can be referred to [1].

The detailed information of the two constructed synthetic SVI databases and MVD database is demonstrated in Table 3. The origin databases and image resolution for generating synthetic SVIs are listed, the contained noise in SVIs and the image numbers in synthetic SVI databases for pre-training are also listed. Similarly, the related information of real MVD dataset is presented.

## 5. Experimental Results and Analysis

In this section, the experimental configuration is first described. The proposed random polygon-based noise for the DIBR distortion simulation is then verified. Afterward, quantitative comparisons among the SVQE models are conducted based on with or without using the SVI datasets generated by the proposed DIBR distortion simulation method. To verify the effectiveness of the proposed scheme by embedding the DIBR distortion mask prediction sub-network into the SVQE models, related experiments are also carried out. The computational complexity comparisons are also described. Finally, the experimental results are discussed.

### 5.1. Experimental Configuration

SVQE models and training setting: Four deep learning-based image denoising or SVQE models, i.e., DnCNN, VRCNN, TSAN, and NAFNet, were employed as testing models. The training scheme is that these models are first pre-trained on synthetic datasets based on NYU-V2/DIV2K and then fine-tuned on the MVD dataset. The common settings for training are that the patch size was set as 128 × 128 and the epoch size was set as 100 for pre-training and 30 for fine-tuning. The batch size was set as 128 for DnCNN and VRCNN and 32 for TSAN and NAFNet. In addition, Adam was adopted as the optimization algorithm with default settings, i.e., $\beta_{1}$ = 0.9, $\beta_{2}$ = 0.999, for DnCNN, VRCNN, and TSAN; AdamW [49] was adopted as the settings, i.e., $\beta_{1}$ = 0.9, $\beta_{2}$ = 0.9, and weight decay $1\;\times \;10^{-5}$, for NAFNet. The initial and minimum learning rates were set as $1\;\times \;10^{-4}$ and $1\;\times \;10^{-6}$ for both DnCNN and VRCNN and $1\;\times \;10^{-3}$ and $1\;\times \;10^{-7}$ for NAFNet while the learning rate was kept the same, i.e., $1\;\times \;10^{-4}$, for TSAN. DnCNN, VRCNN, and NAFNet were trained with the cosine decay strategy, and TSAN kept the default setting, i.e., without using the cosine decay strategy. In addition, the cropped patches for training were randomly horizontally flipped or rotated by 90${}^{\circ }$, 180${}^{\circ }$, and 270${}^{\circ }$. The experiments were conducted on an Ubuntu 20.04.4 operating system with an Intel Xeon Silver 4216 CPU, 64GB memory, an NVIDIA RTX A6000, and a PyTorch platform.

Evaluation metrics: PSNR and IWSSIM [50] are well-known and widely used metrics for conventional 2D images. MPPSNRr [51] and SC-IQA [52] were proposed specifically for DIBR distortion in SVIs and have achieved high correspondence between the predicted quality scores and subjective scores, which may more truly reflect the perceptual quality of synthesized views. In our experiments, both image quality metrics and SVI quality metrics were used to measure the SVI quality.

### 5.2. Verification of Proposed Random Polygon-Based Noise for DIBR Distortion Simulation

In order to verify whether the random irregular polygon-based DIBR distortion generation method is necessary and effective, the database with only compression distortion and the other three different noise patterns (i.e., Gaussian, speckle, and patch shuffle-based noise) as the pre-trained database were employed as the comparison schemes. Note the edge region was located in the same way as that of the proposed method in synthesizing SVIs with other types of random noise. The DnCNN and NAFNet methods were first pre-trained on the NYU database with different distortion schemes and then fine-tuned on the MVD training set.

Table 4 and Table 5 show the denoising performance of DnCNN and NAFNet on the MVD testing sequences Seqs-H.264 among different distortion schemes, respectively. The best and second results for each sequence and on average are highlighted in bold and the best results are underlined again. In terms of both the image quality metrics (i.e., PSNR and SSIM) and the SVI quality metrics (i.e., MPPSNRr and SC-IQA), it can be observed that by pre-training on the NYU database with only the compression distortion could enhance the distorted synthesized video quality on average as compared with the scheme without pre-training. In addition, it could also be found that Gaussian, speckle, patch shuffle-based, and the proposed random irregular polygon-based noise (denoted as randompoly) could contribute to the quality enhancement of the distorted synthesized video. Statistically, by counting the number of occurrences of the best two results of each sequence and on average in Table 4 and Table 5, it could be found that the proposed randompoly noise achieves the best, while the patch shuffle-based performs second. Therefore, it can be inferred that by pre-training on large massive distorted images with different types of noise, the SVQE models could learn more about how to restore images as compared with training on limited MVD data. Our proposed random irregular polygon-based method which could reflect the geometric displacement well is more appropriate to simulate the DIBR distortion, which could greatly elevate the SVQE models’ ability.

To further validate the role of the proposed irregular polygon-based DIBR distortion generation method, a visual quality comparison among different kinds of local noise and SVQE models is performed. Figure 7 and Figure 8 show the quality comparison of sequences Dancer and Poznanhall2 on the pre-training NYU databases with the five different local synthetic noises of the three SVQE models. It can be observed that when only pre-trained on NYU with only compression distortion, the boundaries along the hands and fingers in Dancer and the pillars in Poznanhall2 are clearer than that scheme without pre-training, but they are not as clear as that pre-trained on NYU with other random distortion. By contrast, the SVQE models with the proposed irregular random polygon-based distortion could visually exhibit more pleasant denoised images, which have sharper and complete object boundaries.

### 5.3. Quantitative Comparisons among SVQE Models Pre-Trained with Synthetic Synthesized Image Database

Table 6 and Table 7 demonstrate the denoising/quality enhancement performance of four SVQE models, i.e., DnCNN, VRCNN, TSAN, and NAFNet, on the synthetic synthesized image database (generated from NYU and DIV2K) in terms of image quality metrics and SVI quality metrics. The synthetic databases are termed as SynData for simplicity in the following context. The original model names are used to denote the image denoising/SVQE models only pre-trained on the MVD data, and *model-syn-N/D* is used to denote the image denoising/SVQE models first pre-trained on SynData (NYU/DIV2K) then fine-tuned on the MVD data. Compared to the scheme of four image denoising/SVQE models directly trained on MVD data, it can be observed that four image denoising/SVQE models first pre-trained on SynData then fine-tuned on MVD data can enhance the synthesized views measured by PSNR, IWSSIM, MPPSNRr, and SC-IQA by large gains. Looking at PSNR in Table 6, DnCNN, NAFNet, and TSAN could achieve gains of 0.51-, 0.36-, and 0.26 dB, respectively, while VRCNN could only achieve a gain of 0.08 dB. It is also the same tendency for the four models on the other three metrics. It can be found that the DnCNN and NAFNet models could benefit most from the synthetic dataset on both image quality metrics and SVI metrics. Similar findings can be also observed on DIV2K. In addition, because the images in DIV2K have a 2K resolution, which is similar to that of MVD, and the number of extracted patches from DIV2K is larger than that of NYU, using DIV2K as the pre-trained dataset could have a better performance on average due to the large resolution and larger training samples. The experimental results validate that the proposed SynData with irregular polygon-based distortion could benefit the current SVQE models. In addition, other conclusions could also be drawn. First, a larger synthetic database with the proposed distortion could lead to a better SVQE performance of deep models. Second, different SVQE models would benefit differently with pre-training on the proposed synthetic database.

### 5.4. Effectiveness of Integrating DIBR Distortion Mask Prediction Sub-Network

To further improve the performance of the current SVQE models, the role of the DIBR distortion mask by combining distorted SVIs directly with the ground truth DIBR distortion mask as input to SVQE models was explored and tested. The performance of three DnCNN-based schemes, i.e., DnCNN only trained on MVD (i.e., DnCNN) and DnCNN pre-trained on the NYU (i.e., DnCNN-syn-N) and DIV2K databases (i.e., DnCNN-syn-D), are listed as anchors. Table 8 shows that three corresponding DnCNN-based schemes with ground truth DIBR distortion masks as input, i.e., DnCNN-GTmask, DnCNN-syn-GTmask-N, and DnCNN-syn-GTmask-D, could elevate the distorted synthesized images largely by 0.42-, 0.37-, and 0.39 dB measured by PSNR, respectively. This implies that knowing where the DIBR distortion resides is beneficial to denoise the DIBR distortion.

However, it is actually hard to know the exact position of the DIBR distortion. Thus, similar to those rain or shadow removal works, the detection of the DIBR distortion could be a choice. The noise estimation network used in CBDNet as the DIBR distortion estimation network was employed and then combined with the denoising/SVQE networks to enhance the quality of the outputs. Different from CBDNet, the local DIBR distortion is estimated rather than the whole distortion map. In the experiments, two representative models, DnCNN and NAFNet, were used, and both databases, NYU and DIV2K, were tested. Table 9 shows the average denoising performance on Seqs-H.264 of DnCNN and also NAFNet with and without the DIBR mask prediction network pre-trained on SynData (NYU and DIV2K), respectively. It could be observed that with the DIBR mask estimation network, the quality of the distorted SVIs by DnCNN and NAFNet could be elevated on average in terms of the PSNR, MPPSNRr, and SC-IQA metrics, on both databases, except in IW-SSIM. In addition, the times when the deep models with the DIBR mask prediction perform superior to that without the DIBR mask prediction are counted, and then the surpassing degree is calculated. It can be obtained that the surpassing degrees for the average is 0.69, and for 4/7 of all sequences it is above 0.50, which indicates that our DIBR distortion prediction network works and can further enhance the performance with proposed synthetic databases.

In Figure 7 and Figure 8, it can be observed that part of the DIBR distortion region is repaired while some imprecise repainting is introduced. For instance, the little finger is more clear with the DIBR distortion mask than that without the DIBR distortion mask while some additional noise is introduced along the arm. The reason may lie in that the DIBR distortion prediction network could not precisely predict the DIBR distortion location. Therefore, a more elaborately designed prediction network and architecture are needed for a better SVQE performance.

### 5.5. Computational Complexity Analysis

To test the computational complexity of the proposed random polygon-based (randompoly) DIBR distortion simulation method, experiments on 100 randomly selected images in the NYU Depth Dataset V2 were carried out on a desktop with a windows 10 operating system, an Intel i7-8750H CPU @ 2.20GHz, and the Matlab R2014a platform. As shown in Table 10, compared with other random noise methods, the proposed randompoly method for synthesizing a synthetic SVI takes about 1268.82 milliseconds (ms) on average, while it costs about 60∼70 ms for other random noise generation methods. The computing efficiency shall be further improved for the randompoly method in the future. To test the increased time complexity of introducing a DIBR distortion mask prediction sub-network into an SVQE model, experiments were conducted on the same configurations mentioned in Section 5.1. The calculation time for a frame was averaged over 200 frames for comparison methods. From Figure 9, it can be observed that compared with the original SVQE models, the SVQE models combined with the DIBR distortion mask prediction sub-network consume a little more time. Specifically, the time complexity of introducing the DIBR distortion mask prediction sub-network increases about 1.34 ms (24.53%) and 5.70 ms (3.96%) for a resolution of 1024 × 768 and 4.49 ms (70.12%) and 14.57 ms (3.92%) for a resolution of 1920 × 1088 for DnCNN and NAFNet, respectively, indicating that the computational complexities depend on the complexity of the combined SVQE models and image resolutions.

### 5.6. Discussion

Our proposed random irregular polygon-based (randompoly) DIBR distortion simulation method demonstrates a superior performance than other kinds of random noise in simulating the DIBR distortion. State-of-the-art denoising/SVQE models when pre-training on synthetic SVIs generated by the proposed randompoly method bring large gains in the SVQE performance, objectively and subjectively, than the denoising/SVQE models directly trained on the real MVD dataset. In addition, the proposed DIBR distortion mask prediction sub-network embedded with SVQE models could further enhance the SVQE performance. In the future, a GAN-based or diffusion model-based DIBR simulation method is expected. In addition, more deep investigation is demanded on how to augment images with DIBR distortion and how to effectively introduce the DIBR distortion location information into SVQE models. In addition, transformer-based denoising models for SVQE with synthetic images could be investigated.

## 6. Conclusions

In this paper, a transfer learning-based framework for synthesized image quality enhancement (SVQE) is suggested, in which SVQE models could first be pre-trained on synthetic synthesized images (SVIs) based on substantial RGB/RGBD data, then fine-tuned on real Multi-view Video plus Depth (MVD) dataset, and finally introduce a DIBR distortion mask prediction network together with SVQE models. Different kinds of random noise in simulating DIBR distortion have been explored and validated that the proposed random irregular polygon-based DIBR distortion method is more effective in improving the performance of existing SVQE models. The substantial experimental results on the public MVD sequences demonstrate that existing denoising/SVQE models could achieve large gains by pre-training on synthetic images generated from the proposed random irregular polygon-based method in both image and SVI quality metrics and also demonstrate superior visual quality. In addition, the combination of the DIBR distortion mask prediction network with existing SVQE models has been proved valid for SVQE models.

## Figures and Tables

**Figure 1 sensors-22-08127-f001:**
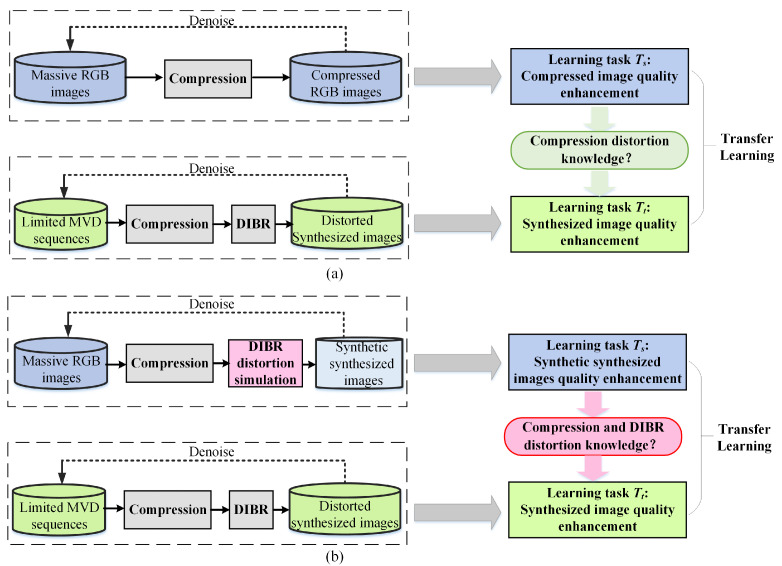
Transferring from learning task $T_{s}$ to task $T_{t}$. (**a**) Compressed image quality enhancement to synthesized image quality enhancement. (**b**) Synthetic synthesized image quality enhancement to synthesized image quality enhancement.

**Figure 2 sensors-22-08127-f002:**
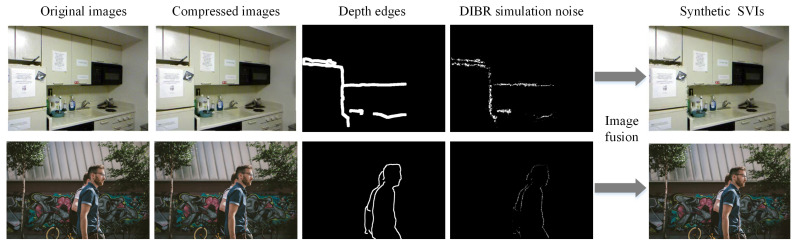
Overview of DIBR distortion simulation pipeline.

**Figure 3 sensors-22-08127-f003:**
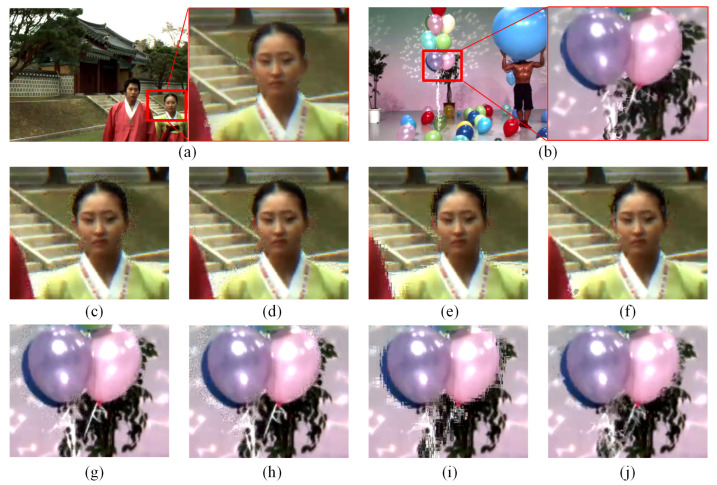
Comparison of DIBR distortion simulation effects by local random noise. (**a**,**b**) are SVIs from sequences Lovebird1 and Balloons, respectively, and the enlarged areas are the representative areas with both compression and DIBR distortion. (**c**–**j**) represent the DIBR distortion simulation effects of rectangle areas in (**a**,**b**) by Gaussian, speckle, patch shuffle-based, and the proposed random irregular polygon-based noise on compressed captured views of Lovebird1 and Balloons, respectively.

**Figure 4 sensors-22-08127-f004:**
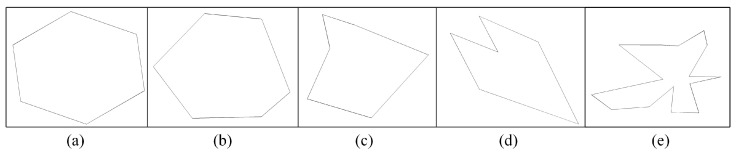
Examples of generated random polygons. *n* denotes the number of vertices, ϵ denotes irregularity, and σ denotes spikiness. (**a**) *n* = 6, ϵ = 0, σ = 0. (**b**) *n* = 6, ϵ = 0.5, σ = 0. (**c**) *n* = 6, ϵ = 0, σ = 0.5. (**d**) *n* = 6, ϵ = 0.7, σ = 0.7. (**e**) *n* = 15, ϵ = 0.7, σ = 0.7.

**Figure 5 sensors-22-08127-f005:**
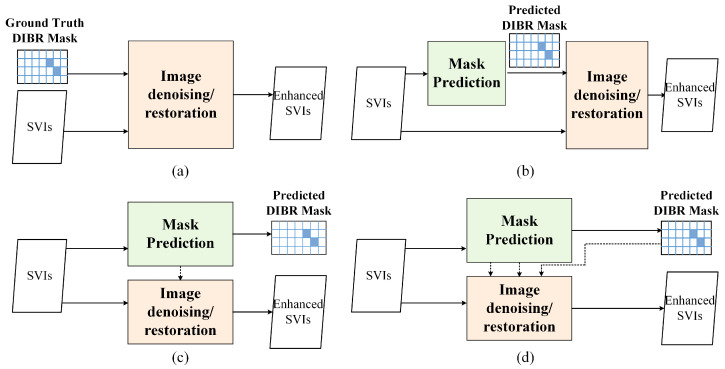
Four possible ways of image denoising/restoration networks integrating with DIBR distortion position. (**a**) Intuitive way of integrating ground truth DIBR distortion position. (**b**) Successive networks with DIBR distortion prediction. (**c**) Parallel networks with DIBR distortion prediction. (**d**) Parallel interactive network with DIBR distortion prediction.

**Figure 6 sensors-22-08127-f006:**
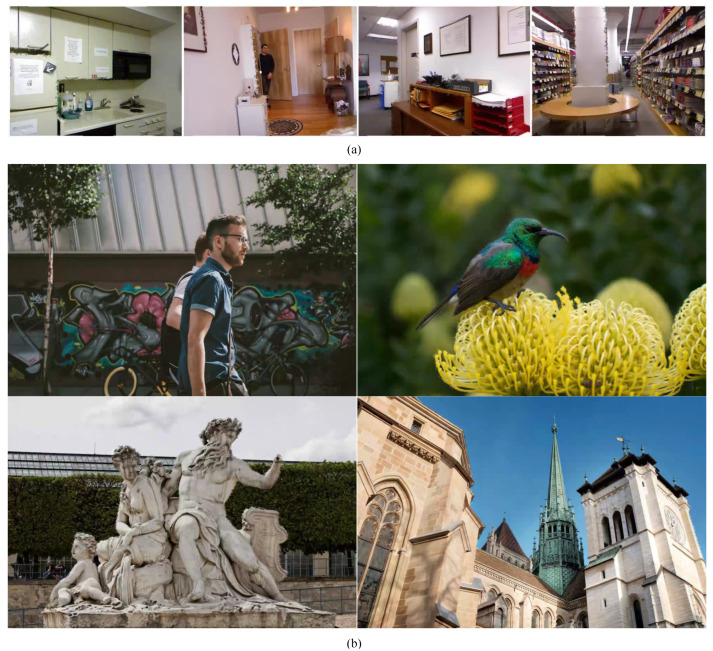
Example images of NYU- and DIV2K-based synthetic image datasets. Zooming for better viewing of synthetic DIBR distortion. (**a**) NYU-based synthetic images. (**b**) DIV2K-based synthetic images.

**Figure 7 sensors-22-08127-f007:**
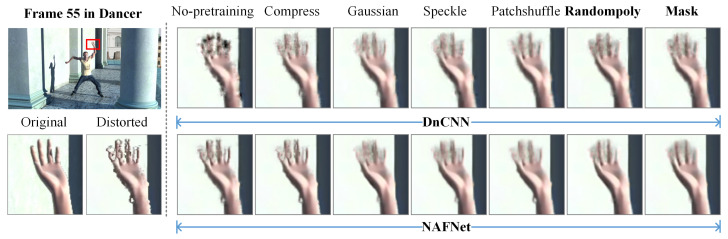
Visual quality comparison of two denoising models, i.e., DnCNN, NAFNet, for SVQE of Dancer with pre-training on synthetic synthesized image database with different random noise types, i.e., compress, Gaussian, speckle, patch shuffle, randompoly (proposed DIBR distortion simulation method), generated from NYU database. ‘Mask’ represents the denoising models that were further integrated with a DIBR distortion mask prediction sub-network using synthetic images generated by ‘randompoly’ method.

**Figure 8 sensors-22-08127-f008:**
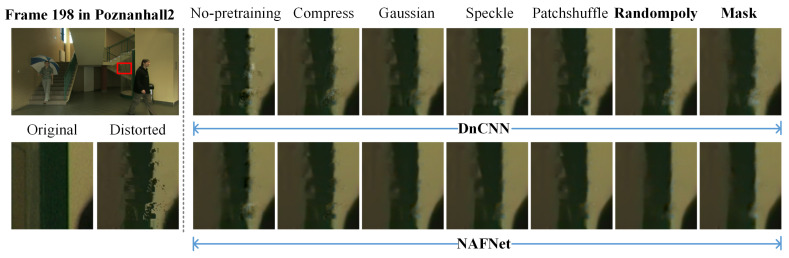
Visual quality comparison of two denoising models, i.e., DnCNN, NAFNet, for SVQE of Poznanhall2 with pre-training on synthetic synthesized image database with different random noise types, i.e., compress, Gaussian, speckle, patch shuffle, randompoly (proposed DIBR distortion simulation method), generated from NYU database. ‘Mask’ represents the denoising models that were further integrated with a DIBR distortion mask prediction sub-network using synthetic images generated by ‘randompoly’ method.

**Figure 9 sensors-22-08127-f009:**
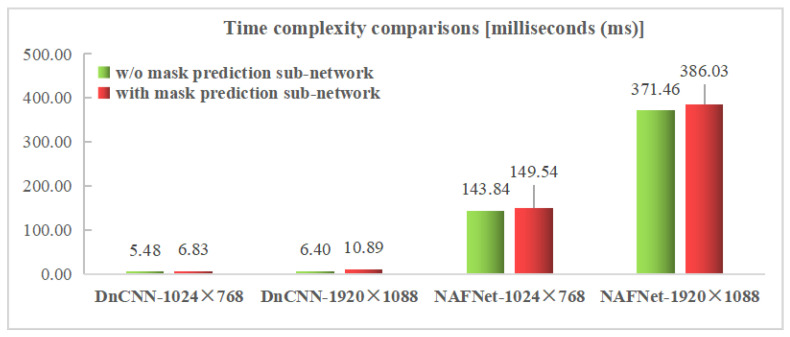
Time complexity comparisons between SVQE models with and without DIBR distortion mask prediction sub-network.

**Table 2 sensors-22-08127-t002:** Definitions of key variables and acronyms.

Variables	Descriptions
$D_{s}$/$D_{t}$, χ, $T_{s}$/$T_{t}$	The source/target domain, feature space, and source/target learning task, respectively
$X=\{x_{1},x_{2},\ldots ,x_{n}\}$	Data samples set, which ∈χ
$D_{s}=\{\left(x_{s_{1}},y_{s_{1}}\right),\left(x_{s_{2}},y_{s_{2}}\right),\ldots ,\left(x_{s_{n}},y_{s_{n}}\right)\},$ $D_{t}=\left\{\left(x_{t_{1}},y_{t_{1}}\right),\left(x_{t_{2}},y_{t_{2}}\right),\ldots ,\left(x_{t_{n}},y_{t_{n}}\right)\right\}$	Ground truth/distorted image pairs for source/target learning tasks
*I*, $I_{\delta }$, $I_{sh}$, $I_{syn}$	A captured view image, *I* added with random noise, *I* added with synthetic geometric distortion generated by the proposed random polygon method, and synthetic synthesized image, respectively
*M*	Mask indicating whether the area in *I* corresponds to strong depth edges
$\theta_{i}$, $r_{i}$, *n*, *R*	The angle, the radius between the *i*-th point and assumed center point, and the number of vertices, the average value of radius of the generated random polygon, respectively.
ϵ, σ	Random variables indicating the irregularity and spikiness of the generated random polygon, respectively
MVD, SVQE, SVI, IQA	Acronyms for Multi-view Video plus Depth, synthesized view quality enhancement, synthesized view image, and image quality assessment, respectively

**Table 3 sensors-22-08127-t003:** Dataset descriptions.

Datasets	Origins/Benchmark	Resolution	Contained Noise	Training	Testing
Synthetic SVI datasets (for pre-training)	NYU Depth Dataset V2 [36]	640 × 480	Compression and synthetic DIBR distortion	1449 images	/
	DIV2K [37]	2K ^1^		750 images	/
Real MVD dataset	SIAT Database [46]	1920 × 1088 /1024 × 768	Compression and DIBR distortion	94 images	1200 images

^1^ The images in DIV2K dataset are of various 2K resolutions, e.g., 1356 × 2040, 2040 × 1536, and 2040 × 960.

**Table 4 sensors-22-08127-t004:** SVQE performance comparison of DnCNN on Sess-H.264 by pre-training on synthetic synthesized image database with different random noise types generated from NYU database. ‘Randompoly’ is the proposed DIBR distortion simulation method and highlighted. The best and second results are highlighted in bold and the best results are underlined again.

Metrics	Models	Kendo	Newspaper	Lovebird1	Poznanhall2	Dancer	Outdoor	Poznancarpark	Average
PSNR	w/o pre-train	33.58	29.79	31.98	34.94	30.90	33.15	30.96	32.19
	Compress	34.05	29.93	32.15	**35.31**	31.89	**33.73**	**31.50**	32.65
	Gaussian	**34.12**	**29.96**	**32.21**	**35.31**	31.97	33.64	**31.51**	32.67
	Speckle	34.09	**29.94**	**32.21**	35.26	31.94	33.66	31.47	32.65
	Patch shuffle	**34.19**	29.88	32.16	**35.31**	**32.16**	**33.75**	**31.50**	**32.71**
	**Randompoly**	34.09	29.93	32.18	35.29	**32.17**	**33.73**	31.49	**32.70**
IW-SSIM	w/o pre-train	0.9318	0.9095	0.9402	0.9067	0.9332	0.9642	0.9215	0.9296
	Compress	**0.9365**	0.9124	0.9420	**0.9108**	0.9421	**0.9679**	**0.9253**	0.9338
	Gaussian	0.9355	0.9132	0.9424	0.9098	0.9433	0.9671	0.9249	0.9338
	Speckle	0.9351	**0.9136**	**0.9427**	0.9087	0.9430	0.9675	0.9252	0.9337
	Patch shuffle	0.9351	**0.9136**	0.9419	0.9094	**0.9448**	0.9675	0.9252	**0.9339**
	**Randompoly**	**0.9357**	0.9132	**0.9425**	**0.9100**	**0.9447**	**0.9678**	**0.9252**	**0.9342**
MPPSNRr	w/o pre-train	36.62	31.53	36.05	37.73	29.40	34.42	34.27	34.29
	Compress	36.98	31.98	36.58	37.82	31.99	**35.10**	34.79	35.03
	Gaussian	37.05	32.07	**36.58**	37.71	32.38	35.02	**34.79**	35.09
	Speckle	37.03	**32.13**	36.54	37.77	32.21	34.91	34.78	35.05
	Patch shuffle	**37.04**	**32.17**	**36.58**	**37.87**	**32.67**	**35.09**	**34.81**	**35.18**
	**Randompoly**	**37.13**	32.10	36.57	**37.91**	**32.66**	34.88	34.79	**35.15**
SC-IQA	w/o pre-train	19.77	17.06	19.32	20.32	15.66	21.86	16.56	18.65
	Compress	20.22	17.55	19.76	20.45	18.01	**24.48**	17.39	19.70
	Gaussian	20.26	17.55	19.88	**20.49**	18.06	23.96	**17.46**	19.67
	Speckle	20.17	17.49	**20.06**	20.43	18.20	24.07	17.37	19.68
	Patch shuffle	**20.29**	**17.55**	19.70	20.49	**18.46**	**24.78**	**17.43**	**19.81**
	**Randompoly**	**20.28**	**17.57**	**19.97**	**20.51**	**18.19**	24.39	17.38	**19.75**

**Table 5 sensors-22-08127-t005:** SVQE performance comparison of NAFNet on Sess-H.264 by pre-training on synthetic synthesized image database with different random noise types generated from NYU database. ‘Randompoly’ is the proposed DIBR distortion simulation method and highlighted. The best and second results are highlighted in bold and the best results are underlined again.

Metrics	Models	Kendo	Newspaper	Lovebird1	Poznanhall2	Dancer	Outdoor	Poznancarpark	Average
PSNR	w/o pre-train	34.00	29.86	32.32	35.39	31.28	33.42	31.50	32.54
	Compress	34.30	30.02	32.30	**35.43**	32.19	**33.90**	**31.66**	32.83
	Gaussian	34.16	29.96	32.27	**35.45**	**32.49**	33.86	31.62	32.83
	Speckle	34.26	29.97	**32.35**	35.40	32.35	33.66	31.63	32.80
	Patch shuffle	**34.29**	**30.07**	32.32	35.40	32.41	33.86	**31.66**	**32.86**
	**Randompoly**	**34.27**	**30.04**	**32.42**	35.41	**32.59**	**33.89**	**31.66**	**32.90**
IW-SSIM	w/o pre-train	0.9386	0.9136	0.9434	0.9151	0.9342	0.9671	0.9245	0.9338
	Compress	**0.9417**	0.9156	0.9444	**0.9158**	0.9454	**0.9694**	**0.9284**	0.9373
	Gaussian	0.9413	**0.9163**	**0.9448**	**0.9166**	**0.9477**	0.9691	0.9279	**0.9377**
	Speckle	0.9414	0.9158	0.9447	0.9156	0.9463	0.9685	0.9272	0.9371
	Patch shuffle	0.9413	0.9159	0.9441	0.9156	0.9468	0.9693	0.9280	0.9373
	**Randompoly**	**0.9416**	**0.9164**	**0.9456**	**0.9158**	**0.9488**	**0.9694**	**0.9285**	**0.9380**
MPPSNRr	w/o pre-train	36.99	32.04	36.57	37.93	32.09	34.76	34.80	35.02
	Compress	37.20	32.20	36.82	**37.97**	32.71	35.41	34.91	35.32
	Gaussian	37.15	32.23	**36.87**	37.95	33.08	35.40	**34.96**	**35.38**
	Speckle	37.22	**32.23**	**36.86**	37.95	32.97	35.44	34.84	35.36
	Patch shuffle	**37.25**	**32.27**	36.85	37.87	**33.14**	**35.51**	34.90	**35.40**
	**Randompoly**	**37.27**	32.11	36.64	**38.04**	**33.11**	**35.48**	**34.96**	35.37
SC-IQA	w/o pre-train	20.04	17.63	20.02	20.59	17.71	23.59	17.28	19.55
	Compress	20.33	17.56	20.06	20.54	18.10	24.20	**17.48**	19.75
	Gaussian	20.35	17.53	**20.36**	**20.60**	**18.64**	24.34	17.36	**19.88**
	Speckle	20.39	17.50	**20.48**	20.59	18.35	**23.53**	17.47	19.76
	Patch shuffle	**20.46**	**17.57**	20.04	20.59	18.43	**24.51**	17.48	19.87
	**Randompoly**	**20.55**	**17.61**	20.34	**20.65**	**18.97**	24.30	**17.50**	**19.99**

**Table 6 sensors-22-08127-t006:** SVQE comparison measured by image quality metrics among DnCNN-, VRCNN-, TSAN-, NAFNet-based schemes by pre-training on synthetic databases from NYU and DIV2K on Seqs-H.264. *Model* (DnCNN, VRCNN, TSAN, NAFNet) represents the baselines, and *model-syn-N/D* represents the existing models (DnCNN, VRCNN, TSAN, NAFNet) combined with transfer learning scheme using our proposed synthetic images. The performance gains of the proposed *model-syn-N/D* over *model* are highlighted in bold.

Metrics	Models	Kendo	Newspaper	Lovebird1	Poznanhall2	Dancer	Outdoor	Poznancarpark	Average
PSNR	DnCNN	33.58	29.79	31.98	34.94	30.90	33.15	30.96	32.19
	DnCNN-syn-N	34.09	29.93	32.18	35.29	32.17	33.73	31.49	32.70 **(+0.51) **
	DnCNN-syn-D	34.15	29.93	32.19	35.30	32.33	33.82	31.52	32.75 **(+0.56)**
	VRCNN	33.90	29.84	32.09	35.14	31.52	33.28	31.36	32.45
	VRCNN-syn-N	33.99	29.87	32.08	35.20	31.59	33.55	31.39	32.52 **(+0.08)**
	VRCNN-syn-D	34.13	29.94	32.12	35.24	32.03	33.75	31.44	32.66 **(+0.21)**
	TSAN	33.93	29.99	32.27	35.03	31.64	33.42	31.08	32.48
	TSAN-syn-N	34.12	29.88	32.16	35.32	32.48	33.84	31.40	32.74 **(+0.26)**
	TSAN-syn-D	34.20	30.04	32.30	35.38	32.45	33.80	31.53	32.81 **(+0.33)**
	NAFNet	34.00	29.86	32.32	35.39	31.28	33.42	31.50	32.54
	NAFNet-syn-N	34.27	30.04	32.42	35.41	32.59	33.89	31.66	32.90 **(+0.36)**
	NAFNet-syn-D	34.40	30.09	32.34	35.51	32.73	34.04	31.71	32.97 **(+0.44)**
IW-SSIM	DnCNN	0.9318	0.9095	0.9402	0.9067	0.9332	0.9642	0.9215	0.9296
	DnCNN-syn-N	0.9357	0.9132	0.9425	0.9100	0.9447	0.9678	0.9252	0.9342 **(+0.0046)**
	DnCNN-syn-D	0.9377	0.9135	0.9436	0.9116	0.9454	0.9681	0.9261	0.9351 **(+0.0056)**
	VRCNN	0.9324	0.9115	0.9406	0.9062	0.9401	0.9662	0.9227	0.9314
	VRCNN-syn-N	0.9337	0.9121	0.9404	0.9068	0.9408	0.9666	0.9234	0.9320 **(+0.0006)**
	VRCNN-syn-D	0.9336	0.9129	0.9410	0.9064	0.9449	0.9675	0.9240	0.9329 **(+0.0015)**
	TSAN	0.9330	0.9138	0.9399	0.9066	0.9407	0.9665	0.9209	0.9316
	TSAN-syn-N	0.9330	0.9138	0.9399	0.9130	0.9471	0.9665	0.9270	0.93433 **(+0.0027)**
	TSAN-syn-D	0.9424	0.9160	0.9445	0.9151	0.9459	0.9686	0.9277	0.93716 **(+0.0055)**
	NAFNet	0.9386	0.9136	0.9434	0.9151	0.9342	0.9671	0.9245	0.9338
	NAFNet-syn-N	0.9416	0.9164	0.9456	0.9158	0.9488	0.9694	0.9285	0.9380 **(+0.0042)**
	NAFNet-syn-D	0.9439	0.9171	0.9452	0.9169	0.9491	0.9702	0.9291	0.9388 **(+0.0050)**

**Table 7 sensors-22-08127-t007:** SVQE comparison measured by SVI metrics among DnCNN-, VRCNN-, TSAN-, NAFNet-based schemes by pre-training on synthetic databases from NYU and DIV2K on Seqs-H.264. *Model* (DnCNN, VRCNN, TSAN, NAFNet) represents the baselines, and *model-syn-N/D* represents the existing models (DnCNN, VRCNN, TSAN, NAFNet) combined with transfer learning scheme using our proposed synthetic images. The performance gains of the proposed *model-syn-N/D* over *model* are highlighted in bold.

Metrics	Models	Kendo	Newspaper	Lovebird1	Poznanhall2	Dancer	Outdoor	Poznancarpark	Average
MPPSNRr	DnCNN	36.62	31.53	36.05	37.73	29.40	34.42	34.27	34.29
	DnCNN-syn-N	37.13	32.10	36.57	37.91	32.66	34.88	34.79	35.15 **(+0.86) **
	DnCNN-syn-D	37.11	31.89	36.51	37.92	33.01	35.24	34.77	35.21 **(+0.92)**
	VRCNN	36.89	32.06	36.54	37.78	31.69	34.69	34.71	34.91
	VRCNN-syn-N	36.97	32.04	36.33	37.84	31.91	34.84	34.77	34.96 **(+0.05)**
	VRCNN-syn-D	36.96	32.22	36.64	37.79	32.89	35.44	34.71	35.24 **(+0.33)**
	TSAN	36.79	32.21	36.58	37.69	32.58	35.29	34.73	35.12
	TSAN-syn-N	37.28	32.23	36.64	37.82	33.39	35.38	34.84	35.37 **(+0.24)**
	TSAN-syn-D	37.26	32.06	36.65	37.89	33.43	35.33	34.87	35.35 **(+0.23)**
	NAFNet	36.99	32.04	36.57	37.93	32.09	34.76	34.80	35.02
	NAFNet-syn-N	37.27	32.11	36.64	38.04	33.11	35.48	34.96	35.37 **(+0.25)**
	NAFNet-syn-D	37.46	32.22	36.83	38.00	33.52	35.55	34.89	35.49 **(+0.47)**
SC-IQA	DnCNN	19.77	17.06	19.32	20.32	15.66	21.86	16.56	18.65
	DnCNN-syn-N	20.28	17.57	19.97	20.51	18.19	24.39	17.38	19.75 **(+1.10)**
	DnCNN-syn-D	20.30	17.58	19.95	20.47	18.31	24.01	17.37	19.71 **(+1.06)**
	VRCNN	20.11	17.46	19.51	20.27	18.14	22.88	17.30	19.38
	VRCNN-syn-N	20.16	17.52	19.47	20.34	17.66	24.12	17.28	19.51 **(+0.13)**
	VRCNN-syn-D	20.14	17.55	19.47	20.32	17.60	24.97	17.16	19.60 **(+0.22)**
	TSAN	19.88	17.59	19.50	20.28	17.34	24.53	16.78	19.42
	TSAN-syn-N	20.33	17.39	19.33	20.52	18.58	24.47	16.89	19.65 **(+0.23)**
	TSAN-syn-D	20.34	17.57	19.75	20.66	18.55	23.72	17.14	19.68 **(+0.26)**
	NAFNet	20.04	17.63	20.02	20.59	17.71	23.59	17.28	19.55
	NAFNet-syn-N	20.55	17.61	20.34	20.65	18.97	24.30	17.50	19.99 **(+0.44)**
	NAFNet-syn-D	20.65	17.60	20.04	20.74	19.15	25.05	17.59	20.12 **(+0.57)**

**Table 8 sensors-22-08127-t008:** SVQE comparison measured by PSNR among DnCNN-based schemes and that with ground truth DIBR distortion masks on Seqs-H.264. DnCNN-syn-GTmask-N, and DnCNN-syn-GTmask-D are abbreviated as DnCNN-syn-GM-N and DnCNN-syn-GM-D, respectively. The performance gains of DnCNN-based schemes with ground truth DIBR distortion masks over DnCNN-based schemes without masks are highlighted in bold.

Models	Kendo	Newspaper	Lovebird1	Poznanhall2	Dancer	Outdoor	Poznancarpark	Average
DnCNN	33.58	29.79	31.98	34.94	30.90	33.15	30.96	32.19
DnCNN-GTmask	34.35	30.13	32.16	34.60	32.61	33.59	30.85	32.61 **(+0.42)**
DnCNN-syn-N	34.09	29.92	32.18	35.30	32.14	33.74	31.50	32.70
DnCNN-syn-GM-N	35.20	30.72	32.44	35.09	33.05	34.00	31.03	33.07 **(+0.37)**
DnCNN-syn-D	34.12	29.91	32.16	35.32	32.29	33.79	31.53	32.73
DnCNN-syn-GM-D	35.35	30.83	32.51	34.92	33.22	34.17	30.86	33.12 **(+0.39)**

**Table 9 sensors-22-08127-t009:** SVQE comparison among DnCNN- and NAFNet-based schemes (*model-syn-N/D*) and that with DIBR distortion prediction (*model-syn-mask-N/D*) on Seqs-H.264. Only when the SVQE models with DIBR distortion prediction (highlighted bold) are superior than the same model pre-trained on the same synthetic database, the results are highlighted bold.

Model	PSNR	IW-SSIM	MPPSNRr	SC-IQA
DnCNN-syn-N	32.70	0.9342	35.148	19.75
**DnCNN-syn-mask-N **	**32.72 **	0.9341	**35.152**	**19.80**
DnCNN-syn-D	32.75	0.9352	35.208	19.71
**DnCNN-syn-mask-D**	**32.78**	0.9345	**35.264**	**19.82**
NAFNet-syn-N	32.90	0.9380	35.372	19.99
**NAFNet-syn-mask-N**	**32.93**	0.9380	**35.493**	**20.00**
NAFNet-syn-D	32.97	0.9388	35.493	20.12
**NAFNet-syn-mask-D**	**32.98**	0.9388	**35.528**	20.08

**Table 10 sensors-22-08127-t010:** The computational complexity comparisons between different types of random noise for DIBR distortion simulation (milliseconds (ms)).

Database	Gaussian	Speckle	Patch shuffle	Randompoly
NYU Depth Dataset V2	67.90	65.61	76.45	1268.82

## Data Availability

Not applicable.

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
