# Peer review of "Deep Learning-Based Synthesized View Quality Enhancement with DIBR Distortion Mask Prediction Using Synthetic Images"

_sensors, 2022, doi:10.3390/s22218127_

Round 1

Reviewer 1 Report

Deep Learning-based Synthesized View Quality Enhancement with DIBR Distortion Mask Prediction Using Synthetic Images

In this work, the authors have proposed a deep learning-based SVQE model using more synthetic Synthesized View Images (SVIs). The paper is well written, and the experimentations are also ok. However, I have the following suggestions.

1.      The abstract should discuss the results acquired based on the proposed methodology. The authors should mention the evaluation metrics and their results. Also, if some comparison is shown in the abstract would be good.

2.      At the end of the introduction section, please write a paragraph to discuss the paper’s organization.

3.      Equations are not cited in the text concerning the equation number.

4.      Dataset is described in the Results sections. Usually, the dataset should be discussed in the methodology section or a separate section.

5.      The authors in the introduction section under the major contribution list say: “A synthetic synthesized view database is constructed,” and in the dataset section, they are referencing to some references. Did they construct the dataset or use the benchmark dataset? Please explain this.

6.      In the dataset section if the authors show some sample images from both datasets. It would be better.

7.      All the dataset details could be shown in pictorial form with a figure; how many images in total? how many to training and testing, labelled, unlabelled etc, etc….

8.      Time complexity should be discussed.

9.      Evaluation metrics should be described in detail in a separate section.

10.   Discussion section is needed wherein the authors should discuss the results.

Author Response

Thank you very much again for your time and effort in providing us with helpful and constructive suggestions to improve the quality of the paper.

The manuscript was mainly revised as follows:

  • Quantitative results performance comparison in terms of different evaluation metrics were added in the abstraction.
  • A paragraph illustrating the paper’s organization was added at the end of the introduction section.
  • The dataset description has been placed in a separate paragraph which was added with a more detailed description. In addition, the figure of sample images in synthetic datasets and a table of information of synthetic and real multi-view video plus depth datasets were added in the revised manuscript.
  • A discussion subsection was added and the evaluation metrics were replaced with a more detailed description in the experimental analysis section.
  • Paragraphs describing the organization of Section 2 and Section 5 were added between the caption of Section 2 and Subsection 2.1, and, Section 5 and Subsection 5.1, respectively.
  • The time complexity comparison was added in the revised manuscript.
  • A nomenclature table that defines variables and acronyms was added.
  • Most first-person sentences were modified as third-person or passive-voice sentences.
  • The language problems and other mistakes were checked and corrected throughout the paper.

Detailed responses to reviewers’ comments are attached below.

Many Thanks!

Yours sincerely

Huan Zhang, Jiangzhong Cao, Dongsheng Zheng, Ximei Yao, Bingo Wing-Kuen Ling

Reviewer 2 Report

The authors present the article entitled “Deep Learning-based Synthesized View Quality Enhancement with DIBR Distortion Mask Prediction Using Synthetic Images”

This paper proposes a deep learning-based SVQE model using more synthetic Synthesized View Images (SVIs). 

The article presents the following concerns:

  • Include quantitative values in the Abstract section in order to highlight the findings.

  • Subsection 2.1: Some techniques are mentioned in this section. I recommend the authors present a table describing the attributions of these works.

  • I suggest adding a Discussion section to present an interpretation of the Results from the perspective of previous studies and the working hypotheses. A comparative table between this work vs. the reported in state-of-the-art is recommended.

  • It is recommendable to include at the end of the introduction a description of the structure of the text.

  • Make a little introduction between points 2 and 2.1, 4 and 4.1, like points 3 and 3.1 

  • Please add a nomenclature table to define variables and acronyms.

  • The text must be written in the 3rd person or passive voice.

  • Avoid using apostrophes and first-person sentences.

The following misspelling should be checked:

  1. lines 3-4: You have written the same word (“multiview”) with and without a hyphen in your document. Both ways are acceptable, but it’s best to be consistent.

  2. line 20: “provide people more and more…” It seems that preposition use may be incorrect here, it’s necessary add “with” before “more”

  3. line 25: “in order to” may be wordy in this sentence, consider changing by “to”

  4. line 33: It appears that “to be able” may be unnecessary in this sentence. Consider removing it. 

  5. line 60:  “pretrained” It appears that you missing a hyphen, consider adding: “pre-trained”

  6. line 65: It appears that the modifiers in the noun phrase “art well-known denoising” are in the wrong order. Consider changing the word order: “well-known art”

  7. line 91: It looks like your sentence contains a redundancy: “mainly transferring” and “distortion mainly” 

  8. line 105: “Bascically” It appears that is a misspelling here

  9. line 105: “categoried” It appears that is a misspelling here, changing by “categorized” 

  10. line 108: “perserve” It appears that is a misspelling here, changing by “preserve” 

  11. line 113: “adaptiation” It appears that is a misspelling here, changing by “adaptation”

Author Response

(The authors gave the same response as above.)

Round 2

Reviewer 1 Report

The authors have revised the paper based on the comments given and I feel it is OK to accept.